# Biomarkers to Predict Lethal Radiation Injury to the Rat Lung

**DOI:** 10.3390/ijms24065627

**Published:** 2023-03-15

**Authors:** Meetha Medhora, Feng Gao, Tracy Gasperetti, Jayashree Narayanan, Heather Himburg, Elizabeth R. Jacobs, Anne V. Clough, Brian L. Fish, Aniko Szabo

**Affiliations:** 1Department of Radiation Oncology, Medical College of Wisconsin, Milwaukee, WI 53226, USA; 2Department of Veterans Affairs, Research Service, Zablocki Veterans’ Administration Medical Center, Milwaukee, WI 53295, USAanne.clough@marquette.edu (A.V.C.); 3College of Dental Medicine, Midwestern University, Downers Grove, IL 60515, USA; 4Cardiovascular Center, Medical College of Wisconsin, Milwaukee, WI 53226, USA; 5Department of Medicine, Medical College of Wisconsin, Milwaukee, WI 53226, USA; 6Department of Mathematics, Statistics and Computer Science, Marquette University, Milwaukee, WI 53233, USA; 7Division of Biostatistics, Institute for Health and Equity, Medical College of Wisconsin, Milwaukee, WI 53226, USA

**Keywords:** microRNA, SPECT/CT, molecular probes, radiation pneumonitis, animal model for radiation injury, biomarkers

## Abstract

Currently, there are no biomarkers to predict lethal lung injury by radiation. Since it is not ethical to irradiate humans, animal models must be used to identify biomarkers. Injury to the female WAG/RijCmcr rat has been well-characterized after exposure to eight doses of whole thorax irradiation: 0-, 5-, 10-, 11-, 12-, 13-, 14- and 15-Gy. End points such as SPECT imaging of the lung using molecular probes, measurement of circulating blood cells and specific miRNA have been shown to change after radiation. Our goal was to use these changes to predict lethal lung injury in the rat model, 2 weeks post-irradiation, before any symptoms manifest and after which a countermeasure can be given to enhance survival. SPECT imaging with ^99m^Tc-MAA identified a decrease in perfusion in the lung after irradiation. A decrease in circulating white blood cells and an increase in five specific miRNAs in whole blood were also tested. Univariate analyses were then conducted on the combined dataset. The results indicated that a combination of percent change in lymphocytes and monocytes, as well as pulmonary perfusion volume could predict survival from radiation to the lungs with 88.5% accuracy (95% confidence intervals of 77.8, 95.3) with a *p*-value of < 0.0001 versus no information rate. This study is one of the first to report a set of minimally invasive endpoints to predict lethal radiation injury in female rats. Lung-specific injury can be visualized by ^99m^Tc-MAA as early as 2 weeks after radiation.

## 1. Introduction

Candidate biological molecules that are altered by radiation have been targets for identification of accurate biomarkers to predict lethal injury [1]. However, despite decades of research, a gap in this field remains. Biological molecules that have been pursued include DNA [2,3], multiple species of RNA such as long non-coding RNA (lncRNA) [4], miRNA [5,6], mRNA [4,7], proteins and protein arrays [8,9,10], advanced multi-omic platforms [11] and others [12]. Essential to bridging this gap is establishing representative animal models and conducting relevant clinical trials. Since it is not ethical to irradiate humans, surrogate animal studies that reflect human injury must be used. The most straightforward model for damage from a radiation attack or accident is total body irradiation (TBI) that induces acute radiation syndrome (ARS). However, in order to induce the delayed effects of acute radiation exposure (DEARE), models that can survive ARS must be developed. By sparing minimal volumes of bone marrow to survive early hematopoietic toxicity, animal models have proceeded to exhibit DEARE [13,14,15]. A part of a single limb of an animal is shielded while exposing the remainder of the total body to ionizing radiation. Such a model is referred to as partial body irradiation with bone marrow shielding (PBI/BM) and has been described in mice [16], rats [13,17] and non-human primates [15,18]. In addition, models that irradiate smaller volumes of the body that spare lethal bone marrow toxicity, for example irradiation to the thorax only (whole thoracic lung irradiation, WTLI [19,20,21]) are used to study the radiotoxicity on single or only a few organs such as the lung and heart [22,23]. Lung injury is also induced in the context of multiple organ exposure by sub-threshold doses of TBI combined with an additional booster dose to the thorax only [8,24].

Regarding the types of radiation used in these models, photon sources such as cesium irradiators [25,26], X-rays [13], linear accelerators [15,18] as well as mixed fields of neutron and X-rays [27] have been applied. Such ionizing radiation manifests ARS and/or DEARE, depending on the dose of radiation delivered and the organs that are exposed. The most common radiation-induced organ injury that has been targeted for biomarker development is radiation to the bone marrow. Kinetics of circulating blood cells have proved very useful to monitor damage represented by this endpoint [12,28,29,30]. As more sophisticated technologies emerge, molecular markers have also become popular [6,7,31,32,33,34,35,36,37]. A few biomarkers for gastrointestinal toxicity during ARS such as circulating levels of citrulline have been described [38,39,40]. In addition, several workshops, reports, and reviews on biomarkers and biodosimetry are available [1,10,41,42,43].

In the current study, we focused on biomarkers for lethal lung injury. The female WAG/RijCmcr rat model was chosen to develop predictive biomarkers for survival from radiation pneumonitis, since well-characterized responses to multiple doses between 0–15 Gy to the thorax (viz: 0-, 5-, 10-, 11-, 12-, 13-, 14- and 15-Gy) were available [13,21,44,45]. A dose of 15 Gy WTLI was the lowest dose previously described to exhibit 100% morbidity from radiation pneumonitis [45] and was therefore hypothesized to manifest the greatest changes by radiation. It was selected to identify candidate biomarkers. Specifically, results of changes in pulmonary perfusion [44], apoptotic injury [44], miRNA expression [46] and circulating blood cell counts [46] were already established at 1, 2, 3 and 4 weeks post-15 Gy WTLI. Our first step was to select the earliest informative timepoint after radiation when biological or molecular alterations could be detected in the lung, but before symptoms of lung injury were observed [47,48]. This approach to initially identify a pulmonary-imaging biomarker was based as rationale to develop a *lung-specific* injury marker after total or partial body radiation exposure. In a mass-casualty event such as a nuclear attack or accident, injury to multiple organs including the heart [49] and lung will occur. The first step to identify pulmonary markers was to use single photon emission computed tomography (SPECT) after WTLI with 15 Gy [44]. The studies were followed by testing other biomarkers including gases in exhaled breath, circulating microRNA (miRNA) in plasma using next-generation miRNA seq and lastly, circulating blood cell counts. Each marker was measured individually in rats at 2 weeks after 15 Gy. Two weeks post-radiation is earlier than the five-week window, when the ACE inhibitor enalapril could be started in the same rat model to mitigate radiation pneumonitis [48]. In addition, it is well within the latent period before any symptoms of radiation pneumonitis are observed [44]. Once the results from this lethal dose (15 Gy WTLI) [44,46], were obtained, we selected two SPECT probes, five candidate miRNAs as well as white blood cells counts for a study on a separate set of rats in the current study. A test group of rats was given a dose of 13 Gy WTLI that was lethal to ≤50% rats (Figure 1). Each rat was then evaluated for all three sets of biomarkers at 2 weeks and then followed to correlate these with the outcome of pneumonitis (lethal or not by 100 days). The same biomarkers were also measured in nonirradiated controls (0 Gy) and in rats irradiated with 10 Gy WTLI [21], doses that did not result in lethal morbidity up to 1 year after irradiation [21]. The collection of biomarkers was limited to a single time point in these cohorts based on maintaining the integrity of the survival studies. The ability of the biomarkers to predict survival was then determined.

## 2. Results

### 2.1. Dose-Response and Survival after WTLI

Figure 1 shows the Kaplan–Meier plots for survival of nonirradiated (0 Gy), 10, or 13 Gy WTLI in female rats. The experiment was terminated by 100 days post-irradiation. While all nonirradiated rats (0 Gy) and 10 Gy survived to 100 days, 50% of rats with 13 Gy were morbid by 60 days. All rats surviving to 60 days continued to stay alive until the study was terminated. Previous studies had determined that there were no survivors at 60 days after 15 Gy WTLI in the same model [45]. In addition, rats given 13 Gy WTLI that survived pneumonitis went on to survive over 210 days [13].

### 2.2. Perfusion Volume of the Lung and Apoptosis as Measured by SPECT/CT Imaging at 2 Weeks Post-Irradiation

The fraction of the lung volume that was perfused in each rat was determined at 2 weeks following radiation using technetium-labeled macroaggregated albumin (^99m^Tc-MAA) and SPECT imaging. The ^99m^Tc-MAA beads (diameter < 20 μm) lodge in capillaries in proportion to flow. Differences between perfused volumes for SPECT studies between nonirradiated controls versus rats irradiated with 10 Gy or 13 Gy WTLI are shown in Figure 2 and were analyzed by a Kruskal–Wallis test with Dunn’s test to account for multiple comparisons. There was no statistical difference between perfusion in nonirradiated rats, median 93.9% with 95% confidence intervals of (64.1–96.8, n = 6) versus those given 10 Gy WTLI median 92.8% (73.5–98.6, n = 7) or 13 Gy WTLI median of 87.3% (70.9–91.6, n = 26). A Mann–Whitney test was performed comparing nonirradiated controls with a median of 93.9% (64.1–96.8, n = 6) to rats that were morbid by 60 days after 13 Gy WTLI with a median of 73.5% (62.0–91.1, n = 13), these results reached statistical significance (*p* = 0.037) and are comparable to the 15 Gy WTLI rat cohort (n = 11) in which matched nonirradiated controls (n = 12) showed a significant decrease in perfusion of the irradiated lungs [44].

Values for uptake of the apoptosis marker technetium-labeled duramycin (^99m^Tc-DU, that binds to apoptotic cells) were not different after 10- or 13-Gy and are not included in Figure 2 but are shown in Appendix A where all results are graphically summarized. It should be noted that uptake of both probes, ^99m^Tc-MAA and ^99m^Tc-DU, is known to be altered in the lung at 2 weeks after a lethal dose of 15 Gy WTLI [44].

### 2.3. Changes in Circulating White Blood Cell Counts at 2 Weeks after Irradiation

Figure 3 shows white blood cell counts at 2 weeks after 10- and 13-Gy WTLI. Total white blood cells were decreased at 2 weeks after both doses (Figure 3A), similar to previous results for 15 Gy WTLI [46]. The percent of neutrophils in the white blood cell compartment were increased by 13 Gy WTLI but not after 10 Gy WTLI (Figure 3B). The percent of lymphocytes in the white blood cells were decreased by 13 Gy WTLI but not after 10 Gy WTLI (Figure 3C). The percent of monocytes in white blood cells tended to be higher after 13 Gy WLTI (Figure 3D), though this difference did not reach statistical significance. Because differences in any one 2-week variable alone did not correlate to survival, we proceeded to test a combination of endpoints. It should be noted that the absolute differential counts of the individual white blood cell types were not changed; only the percentages of some cell types were altered as shown in Figure 3B,C. The absolute counts of individual white blood cell types are not included in Figure 3 but are shown in Appendix A where all results are graphically summarized.

### 2.4. Expression of miR19a-3p, miR142-5p, miR 144-5p, miR 144-3p, miR 21-5p at 2 Weeks after Irradiation

The results of the relative expression of miR 19a-3p, miR 142-5p, miR 144-5p, miR 144-3p, miR 21-5p and 150-5p after normalization to miR191-5p are shown in Figure 4. miR191-5p was selected for normalization as we have demonstrated it is not altered by radiation [46]. The relative expression of miR 144-5p, miR 144-3p, miR 142-5p, miR 19a-3p and miR 21-5p (markers that were upregulated in whole blood compared to miR 191-5p at 2 weeks after 15 Gy WTLI [46]) were not significantly increased by 13 Gy WTLI (Figure 4A–E). MiR 150-5p, a marker derived from hematopoietic cells [6] was decreased after 13 Gy WTLI (Figure 4F). Similar results were obtained after 10 Gy WTLI.

### 2.5. Combining Biomarkers to Predict Lethal Lung Irradiation

#### 2.5.1. Data Import

Data were available from two sets of experiments. The first set had all variables of interest measured in animals treated with 0-, 10- or 13-Gy radiation, while the second set had only one type of variable (i.e., only blood composition, only miRNA, only ^99m^Tc-MAA or only ^99m^Tc-DU) in animals treated with 0- or 15-Gy. Univariate analyses were conducted on the entire combined dataset, using the animals with non-missing values of the relevant variable (See Appendix A). The multivariate analyses were conducted using an artificial dataset with imputed values for the single-variable animals from the second set as described below.

#### 2.5.2. Imputation Procedure

Based on the four subsets depending on the variable availability, 29,898 hypothetical animals were created with all possible combinations of the values for the variables, matching on radiation dose. From this pool, 11 animals treated with 0 Gy and 11 animals treated with 15 Gy were randomly selected to reflect the typical group sizes in the second set of experiments. Six 0 Gy, all the 10 Gy and 13 Gy animals had complete data, and they were included as is (see Table 1). This process can be considered a single replicate of a multiple imputation procedure. A limitation of this approach is the assumption of independence between the variables, conditional on radiation dose, during the imputation process.

#### 2.5.3. Predicting Survival

Prediction rules for 60-day status were then developed using a classification-tree (CART) approach. CART uses a recursive binary splitting algorithm to build the classification tree. Using the entire dataset, the cutoff for the optimal split for each variable is determined by minimizing the Gini index, which measures the “impurity” of the nodes created by the split. The variable with the lowest Gini index is selected for the split. This process is repeated in the two subsets defined by the first split, and all subsequent splits until there is no improvement in the classification or the subsets become too small (we did not split nodes with fewer than 20 observations). The benefits of CART include its interpretability, ability to model interactions and ability to handle missing data. Splitting criteria are listed at each decision point; the left branch corresponds to “YES” and the right branch to “NO”. The Alive/Dead label at the bottom of the tree (Figure 5) indicates the classification of the observations falling into that group. Missing values were handled by either splitting on a correlated “surrogate” variable or by going with the majority. Figure 5 shows that a combination of percent lymphocytes and monocytes, as well as a drop in pulmonary perfusion volume will predict survival from radiation to the lungs with 88.5% accuracy (95% confidence interval of 77.8, 95.3) with a *p*-value of < 0.001 versus the no information rate (NIR) of 60.7%, and a McNemar’s *p*-value of 0.45. All the misclassified observations were in the 13 Gy group, within which the accuracy was 73.1% (95% CI: 52.2–88.4%), with *p* = 0.015 versus NIR.

## 3. Discussion

The goal of the current study was to find markers to identify potentially lethal radiation-induced pneumonitis. Since multiple organs are injured by radiation and circulating markers such as blood cells and miRNA can be altered by radiation to organs other than the lungs, pulmonary imaging modalities are likely to directly measure lung injury. SPECT imaging with ^99m^Tc-MAA detected changes as early as 2 weeks in rats, providing a structural biomarker of perfused lung volume that correlates with lung function, since perfusion plays an important role in gas exchange in the lung. There are many reports of a decrease in perfusion in the lungs after radiation [50,51,52]. In the future, SPECT/CT values in nonirradiated humans will be useful to provide a baseline for this biomarker to make it informative after radiation. Another change at 2 weeks after radiation, vascular permeability, has also been reported [53]. This dysfunction has been explored using an FDA-approved dye in combination with optical imaging in rat lungs [53], a technique that could be easily applied in the clinic [54,55]. These translational markers can readily be measured in humans. Differential white blood cell counts can be easily evaluated at 2 weeks, while clinics throughout the United States routinely measure lung perfusion by SPECT/CT. It should be noted that there are only a few other studies measuring such early functional changes in any organ to predict radiation injury. One example is cardiac magnetic resonance imaging for evaluation of cardiotoxicity in patients with breast cancer [56]. We report that a combination of percent change in lymphocytes and monocytes, as well as pulmonary perfusion volume, could predict survival from radiation to the lungs with 88.5% accuracy (95% confidence intervals of 77.8, 95.3) with a *p*-value of <0.0001 versus no information rate. Thus, a simple set of tests has potential to detect lethal lung injury before pulmonary symptoms of radiation pneumonitis develop and in time to mitigate injury with an ACE inhibitor [48].

Blood cell measurements, ratios and kinetics are commonly used as diagnostic biomarkers for radiation exposure [30] and have been refined for dose evaluation [29] and biodosimetry [28]. They have been combined with other markers to assess multiple parameters for radiation injury in nonhuman primates [28]. RNA and gene expression have also been popular candidates as radiation biomarkers. Some of the more recent and successful examples have been the measurement of miR 150-5p [6] as well as expression of radiation responsive genes [7,10]. RNA signatures have also been explored in nonhuman primates exposed to total body irradiation and WTLI [5]. Emerging as another set of popular candidates are small molecules such as metabolites and lipids [34,35,49]. Other investigators have also attempted to identify radiation-induced biomarkers of lung injury in animal models relevant to a radiological incident [5,8,43]. Circulating content of club cell secretory protein (CCSP) and surfactant protein D (SP-D) expressed by lung epithelium were identified as biomarkers for pulmonary fibrosis induced by radiation [8].

Since circulating blood cell counts can be altered by a number of factors, we also evaluated if intraperitoneal injection of a toxic agent, lipopolysaccharide (LPS), alters circulating blood cell counts in a similar manner as radiation to the lungs. After 24 h treatment with LPS, a time point that injures the lungs, the neutrophil/lymphocyte ratio was 3 times higher than with radiation at 2 weeks after >10 Gy WTLI (results not shown). Thus, the profile or even kinetics of changes in circulating blood cell counts may differentiate between these two insults (biochemical and radiological toxicities). Though the pulmonary SPECT endpoint advances the search for a lung-specific biomarker, further research by irradiating other parts of the body and shielding the lungs will verify if perfusion is altered even when the lung is not in the radiation field. In addition, evaluating male rats will determine if the biomarkers are sex-specific. Though the pattern and effects of radiation to the lungs are very similar after WTLI and PBI with bone marrow sparing [57,58], the biomarkers presented in this study were not tested after PBI. Another limitation of the current work is the use of the WAG/RijCmcr rat model. Humanized animal models [59] are gaining popularity and can better confirm translational potential of biomarkers studied in rodents. Males and special populations such as juvenile and geriatric models must also be studied in the future to determine efficacy of the biomarkers described here. In addition, as mentioned, baseline ^99m^Tc-MAA SPECT/CT values in human lungs will help to promote this biomarker of lung perfusion to detect lethal radiation injury.

In summary, the current study is one of the first to report a set of non-invasive structural and functional endpoints that are translational and can be used to predict lethal radiation injury in female rats in combination with simple tests such as circulating blood cell counts. Detection is possible as early as 2 weeks after radiation, within the latent window before symptoms of radiation injury to the lung are observed. Mitigators can then be used after detection to reduce pulmonary toxicity induced by radiation to the lungs.

## 4. Materials and Methods

### 4.1. Animal Care

All animal protocols were approved by Institutional Animal Care and Use Committees (IACUC) at the Medical College of Wisconsin [23].

### 4.2. Irradiation and Follow Up

Unanesthetized female WAG/RijCmcr rats were irradiated with 10 Gy (n = 7) or 13 Gy (n = 26) WTLI at 9–10 weeks of age. Nonirradiated age-matched control rats (0 Gy) (n = 6) were also evaluated. WTLI dosimetry was conducted as has been previously described [47]. Briefly, rats were placed in a Plexiglas jig to allow for irradiations to be performed without anesthesia. Rats were exposed with opposed parallel fields that encompassed the whole thorax with an X-RAD 320 X-ray unit (Precision X-Ray, Madison, WI, USA) at 320 kVp and 13 mA at a dose rate of 1.531 Gy min^−1^ with a half value layer of 1.4 mm Cu filtration. Biomarkers were studied at 2 weeks after WTLI. This time point was determined by results from biomarkers measured in rats given 15 Gy WTLI at 4 time points in an earlier study [44].

Irradiated rats (13 Gy WTLI and 10 Gy WTLI) and controls (0 Gy) were followed to 100 days to determine survival. Nonirradiated rats of the same age, strain and gender served as controls in parallel with each batch of irradiated rats. Animals that were evaluated as morbid were euthanized as in previous studies [17,23].

### 4.3. SPECT/CT

In vivo SPECT/CT was performed as described previously [44] using 2 probes: technetium-labeled macroaggregated albumin (^99m^Tc-MAA) and technetium-labeled duramycin (^99m^Tc-DU). ^99m^Tc-MAA lodges within the intricate and abundant microvasculature of the lung in proportion to flow, and gamma-ray emissions are detected by the gamma camera in the SPECT imager. ^99m^Tc-DU has high affinity and specificity for phosphatidylethanolamine (PE) which is externalized in apoptotic and other dying cells [44]. Since radiation has been reported to induce apoptosis [60,61,62], necrosis and mitotic cell death, this marker reports whole body tissue damage after radiation [63].

Macroaggregated albumin (Jubilant DraxImage) and duramycin (3035 g/mole MW) were labeled with technetium as previously described [44,63,64,65]. Rats first received an injection (tail vein) of 38.5 ± 7.6 (mean ± 95% confidence intervals) MBq of ^99m^Tc-duramycin. The animals were positioned in a SPECT/CT scanner (Triumph, TriFoil Imaging) and rapidly CT scanned for anatomical localization then scanned at 50 min after injection. Rats were then injected with ^99m^Tc-MAA administered as a single dose of 24.6 ± 3.9 (mean + Standard Deviation (SD)) MBq via the tail vein, using the same catheter. A second scan was run at 5 min after the ^99m^Tc-MAA injection. The corresponding maximal uptake for each probe was determined as in previous studies. In vivo radionuclide imaging was performed using multi-pinhole collimators with two gamma head detectors (130 to 150 keV energy window). A total of 72 projections for 10 s each were acquired for each scan. SPECT and CT data were reconstructed and coregistered using built-in software [44,63]. CT scans were performed again on each rat after SPECT imaging to determine lung volume.

### 4.4. Image Analysis

The reconstructed CT and ^99m^Tc-MAA image volumes were segmented and analyzed to determine the perfused lung volume as previously described [44]. Briefly, the lung region within the CT image volume was identified. A grayscale window with a lower threshold of zero (corresponding to air) and an upper threshold of the maximum grayscale value within the lung parenchymal region, was established and used to determine the boundaries of the anatomical lung region. This region served as a binary lung mask that was then applied to the reconstructed radiolabeled image volumes.

For the ^99m^Tc-MAA studies, the total number of nonzero voxels of ^99m^Tc-MAA SPECT lung region for each rat was determined and scaled to the volume of each SPECT voxel, to determine the perfused lung volume. The fraction of lung perfused was determined by dividing the perfused lung volume obtained from ^99m^Tc-MAA by the anatomical lung volume obtained from CT.

For experiments involving ^99m^Tc-DU, the reconstructed CT and SPECT image volumes were coregistered and lung boundaries from CT were applied as a mask to determine the ^99m^Tc-DU lung region. ^99m^Tc-DU lung uptake was then determined as average ^99m^Tc-DU counts per voxel within the lung region and normalized to the activity of ^99m^Tc-DU injected [44].

### 4.5. Blood Cell Counts

Rats were anesthetized for blood-draws (0.5–1 mL) by retro-orbital bleeds conducted by an experienced technician [66]. Total and differential white blood cell counts were obtained the same day by Marshfield Laboratories (Marshfield, WI, USA) [46]. Only samples with adequate volume of uncoagulated blood were considered for evaluation.

### 4.6. Measurements of Expression of Circulating microRNA

Candidate microRNA, miR 144-5p, miR 144-3p, miR 142-5p, miR 19a-3p and miR 21-5p that were upregulated after 15 Gy WTLI in a previous study [46] were tested as biomarkers, along with miR 150-5p that was downregulated and miR 191-5p that was not changed after 15 Gy WTLI [46]. In the current study, total RNA was prepared from whole blood from control rats (0 Gy) and rats irradiated to 10 Gy or 13 Gy WTLI at 2 weeks after WTLI by the same method described for rats given 15 Gy WTLI [46]. Briefly, RNA was isolated from whole blood using TRIzol Reagent (Ambion/RNA (Life Technologies, Carlsbad, CA, USA). High purity total RNA (A260/280 > 1.6) was used for RT-qPCR without pre-amplification. LNA-primers, RT-qPCR and calculation of miRNA were performed as described [46,67]. The expression of each target that was upregulated: miR 144-5p, miR 144-3p, miR142-5p, miR19a-3p and miR 21-5p was separately compared to the expressed level of miR 191-5p in the same blood sample.

### 4.7. Statistical Analyses

Kruskal–Wallis test with Dunn’s test for multiple comparisons were used to test significance for lung perfusion, blood counts and miRNA. Median values with 95% confidence intervals are presented.

### 4.8. Strategy for Identifying and Testing Biomarkers

Candidate biomarkers from previous results from rats exposed to 15 Gy along with nonirradiated controls [44,46] were used to serve as a base to narrow down an informative set of biomarkers to predict lethal radiation injury to the lungs. SPECT imaging had yielded 2 biomarkers, Tc-MAA and Tc-DU, that determined the volume of perfused lung as well as apoptosis in the lung respectively after 15 Gy WTLI as published [44]. miRNA-seq at 2 weeks had yielded 5 plasma miRNA that were upregulated after 15 Gy WTLI [46]. A third set of rats given 15 Gy WTLI along with nonirradiated controls was used to measure total and differential blood cell counts at the same time point, 2 weeks after 15 Gy WTLI. In the current study, all 3 sets of biomarkers were measured in the same rat at 2 weeks after 13 Gy WTLI, a radiation dose which had been characterized to cause lethal pneumonitis to ~50–70% of irradiated rats [45,47]. After biomarkers were measured at 2 weeks with 13 Gy WTLI, the same rats were followed to 100 days to determine survival of each animal and then correlate the biomarker results with survival. The 3 sets of biomarkers (SPECT probes, miRNA and blood cells counts) were also evaluated at 2 weeks after radiation in nonirradiated controls (0 Gy) and rats given 10 Gy WTLI, a dose that is survived by all rats. In previous studies, rats surviving to 100 days went on to survive at least 210 days and 1 year after irradiation [47,57]. Statistical methods were then developed to define a biomarker signature to predict survival after radiation as well as determine the sensitivity and specificity of the selected biomarker panel.

## Figures and Tables

**Figure 1 ijms-24-05627-f001:**
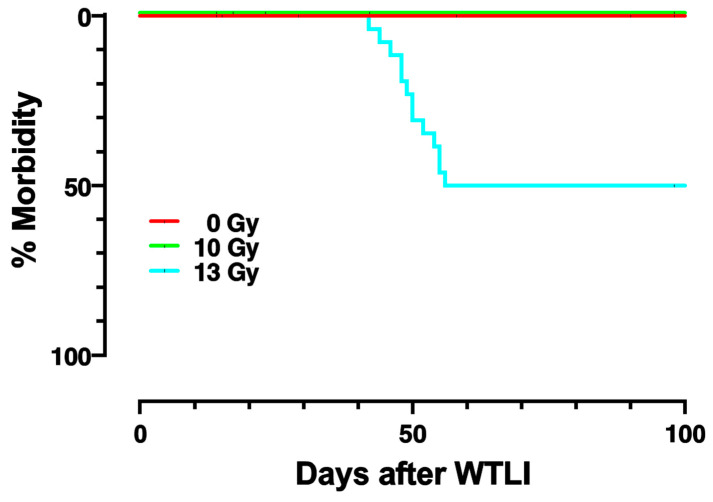
Kaplan–Meier plots for percent morbidity (X-axis) from pneumonitis in control female rats and those given whole thorax lung irradiation (WTLI) and followed for 100 days post-irradiation (X-axis). Three doses of radiation are represented: (1) no irradiation (0 Gy—red line), (2) 10 Gy WTLI (green line) and (3) 13 Gy WTLI (blue line).

**Figure 2 ijms-24-05627-f002:**
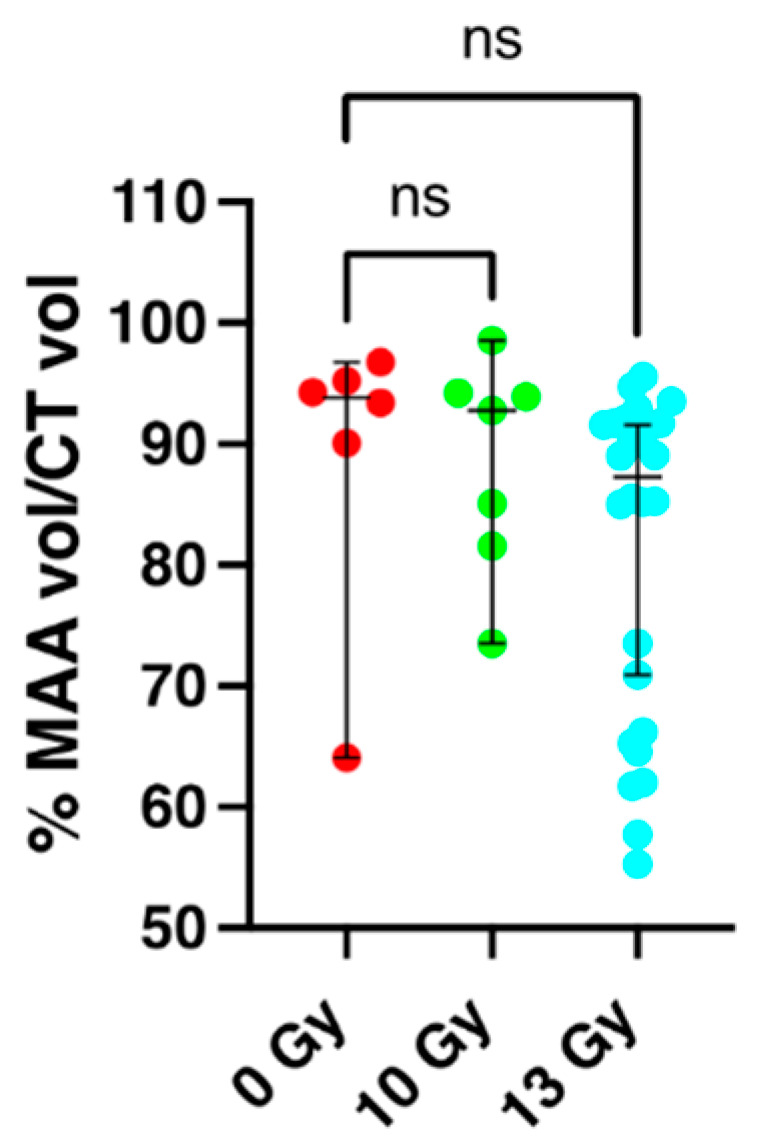
Percent perfused lung volume by radiation (Y-axis) as determined by ^99m^Tc-MAA and CT imaging. The data are presented as medians and 95% confidence intervals in female nonirradiated rats and age-matched rats given 10- and 13-Gy whole thorax lung irradiation (WTLI) at 2 weeks after exposure. There was no difference in perfusion between groups when all rats in each group were included. ns = no statistical difference.

**Figure 3 ijms-24-05627-f003:**
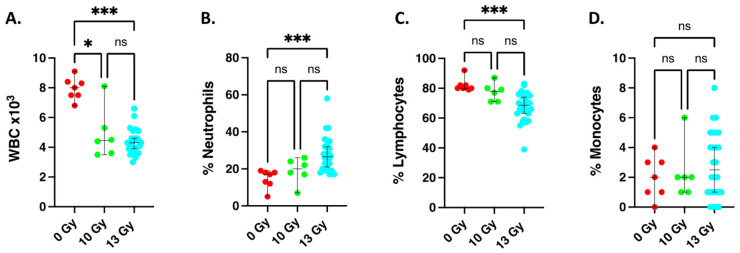
Graphical representation of circulating white blood cell counts 2 weeks after whole thorax lung irradiation (WTLI). Blood was harvested from female nonirradiated, and age-matched rats given 10- and 13-Gy WTLI and the white cell count determined as described (see Section 4). (**A**) Total white blood cells/microliter. (**B**) Percent neutrophils. (**C**) Percent lymphocytes. (**D**) Percent monocytes. The data are presented as medians and 95% confidence intervals: * = *p* < 0.0332, *** = *p* < 0.0002, ns = no statistical difference.

**Figure 4 ijms-24-05627-f004:**
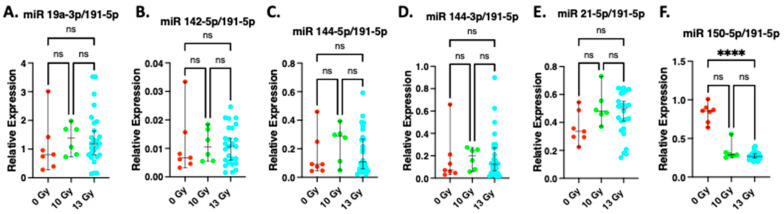
miRNA expression relative to miRNA 191-5p: Graphical representation of circulating miRNA from total blood after whole thorax lung irradiation (WTLI). Blood was harvested from female nonirradiated (0 Gy) and age-matched rats given 10- and 13-Gy WTLI and miRNA isolated and quantitated by qRT-PCR as described (see Materials and Methods). Normalized expression of miRNA as compared to miR 191-5p: (**A**) miR 19a-3p, (**B**) miR 142-5p, (**C**) miR 144-5p, (**D**) miR 144-3p, (**E**) miR 21-5p, (**F**) miR 150-5p. The data are presented as medians and 95% confidence intervals: **** = *p* < 0.0001, ns = no statistical difference.

**Figure 5 ijms-24-05627-f005:**
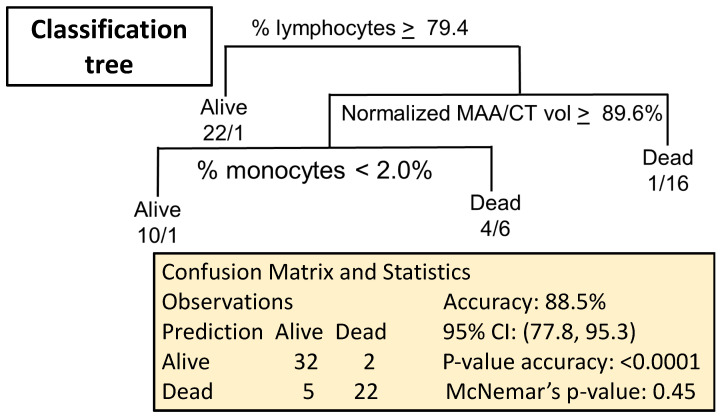
Prediction rules for 60-day status were developed using a classification-tree (CART) approach. In the plots, the splitting criterion is listed at each decision point; the left branch corresponds to “YES” and the right branch to the “NO”. The Alive/Dead label at the bottom of the tree indicates the classification of the observations falling into that group, with the numbers indicating the actual number of observations (alive/dead). Splits on a missing value are handled by either splitting on a correlated “surrogate” variable, or by going with the majority. McNemar’s *p*-value: test misclassification bias (e.g., more alive > dead or dead > alive).

**Table 1 ijms-24-05627-t001:** Number and distribution of rats used for biomarker analysis.

Number	Gy	Notes
6	0	Study set
7	10	Study set
26	13	Study set
11	0	Biomarker identification set
11	15	Biomarker identification set

## Data Availability

The data supporting findings presented in this study are available from the corresponding author upon reasonable request.

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
