# Peer review of "Biomarkers to Predict Lethal Radiation Injury to the Rat Lung"

_ijms, 2023, doi:10.3390/ijms24065627_

Round 1

Reviewer 1 Report

The study under review reports a set of blood biomarkers and lung perfusion that predict lethal radiation injury in female rats. By analyzing changes in white blood cells, lung perfusion and miRNA levels it is concluded that a combination of percent change in lymphocytes and monocytes and lung perfusion volume could predict survival from radiation injury to the lung.

Comments:

1.     Figures 2 and 3 show that lung perfusion and percent monocytes are not altered significantly in rats exposed to 13 Gy radiation compared to control rats, yet it is claimed that their changes predict survival from radiation injury. The data contradict the conclusions.

2.     Figure 5 - What is the rationale for normalization of miRNA levels to miR 150-5p? As miR 150-5p levels are significantly reduced upon radiation exposure, the levels of other miRNAs measured and compared to control (0 Gy) rats are bound to show increased levels.

3.     The results of imputation method (Figures 6 and 7) to predict survival and the classification tree (Figure 8) are not comprehensible to this reviewer.

4.     The discussion section contains quite a bit of recap of results that should be removed from the text.

Author Response

We would like to thank the reviewer for your time and efforts in reviewing our manuscript entitled “Biomarkers to predict lethal radiation injury to the rat lung”.  In response to your suggestions and comments we have made multiple revisions to the manuscript, expanding the analysis and removing results from discussion section, and modifying the figures. We feel these changes have improved the manuscript and we sincerely hope you find it suitable for publication.

Reviewer 1 Open Review

Comments and Suggestions for Authors

The study under review reports a set of blood biomarkers and lung perfusion that predict lethal radiation injury in female rats. By analyzing changes in white blood cells, lung perfusion and miRNA levels it is concluded that a combination of percent change in lymphocytes and monocytes and lung perfusion volume could predict survival from radiation injury to the lung.

Comments:

  1. Figures 2 and 3 show that lung perfusion and percent monocytes are not altered significantly in rats exposed to 13 Gy radiation compared to control rats, yet it is claimed that their changes predict survival from radiation injury. The data contradict the conclusions. Our explanation of the chosen approach was not optimal. Because reliance on a single variable at a time was not informative, we assessed a joint distribution. Our model suggests that death occurred mostly when both lymphocyte percent and lung perfusion decreased. Combined but not individual endpoints can predict mortality. See page 4, section 2.3 in the manuscript for changes in the text.

  1. Figure 5 - What is the rationale for normalization of miRNA levels to miR 150-5p? As miR 150-5p levels are significantly reduced upon radiation exposure, the levels of other miRNAs measured and compared to control (0 Gy) rats are bound to show increased levels.

As reviewer points out, the miR150-5p is decreasing after radiation and this change is in a different direction from all the other five microRNAs. We used the ratio of the five increased microRNAs to miR150-5p for a combination expression that involves changes from both directions. This combined expression ratio increases the sensitivity of miRNA to detect radiation evoked changes. It also minimizes human error due to pipetting, etc. However, it must be clearly stated that ratios (figure 5) combine changes from both microRNAs, and therefore should not be treated statistically or conceptually as an expression of the single microRNA. To address the reviewer’s point, we have modified the manuscript in session 2.4: “The ratio of the expression of each of the 5 miRNAs to miR 150-5p was also calculated for each marker (Figures 5A-E) since it would show more pronounced change because of the marked decrease in expression of miR 150-5p. This ratio was higher for miR19a-3p, miR142-5p, miR144-5p and miR 21-5p after 13 Gy WTLI. The ratio of expression of miR144-5p and miR 21-5p derived by comparison to miR 150-5p were also increased at 2 weeks after 10 Gy WTLI (Figure 5E).” as following “The ratio of the expression of each of the 5 miRNAs to miR 150-5p was also calculated for each marker (Figures 5A-E). The products were shown as a relative expression, which is a combination of two oppositely changed microRNA to increase sensitivity of radiation evoked changes in miRNA expression and to normalize for variability based on pipetting, etc. This ratio was higher for miR19a-3p/150-5p, miR142-5p/150-5p, miR144-5p/150-5p and miR21-5p/150-5p after 13 Gy WTLI. The ratio of expression of miR144-5p/150-5p and miR 21-5p/150-5p were also increased at 2 weeks after 10 Gy WTLI (Figure 5E). Page 5 section 2.4

3.The results of imputation method (Figures 6 and 7) to predict survival and the classification tree (Figure 8) are not comprehensible to this reviewer.

Figures 6 and 7 do not show the results of the imputation. They are two complementary visualizations of the univariate relationships between the survival and the collected variables across all radiation doses. Additional explanation about the classification tree has been added to the manuscript in sections 2.5.1 and 2.5.2.

  1. The discussion section contains quite a bit of recap of results that should be removed from the text. Thank you. Reiterated results have been removed from the discussion.

    In closing, the authors appreciate the efforts of the reviewer and editors’ suggestions that improved this manuscript.

    Thank you again.

Reviewer 2 Report

General comments

The manuscript reports the results of the study that aimed to investigate and establish effective biomarkers of lethal lung injury after exposure to ionising radiation. An ultimate objective of the study is to predict lethal outcome at early stages before manifestation of any symptoms thus allowing to apply countermeasures to improve survival. The subject of investigation presents a great interest for a broad audience of researchers, and its importance is undoubtful. The study is based on the extensive pool of experimental data obtained using relevant range of techniques. The authors claim the establishment of a combination of three biomarkers and the use of the classification tree approach for the prediction of lethal outcome. There are however a few issues that undermine the interpretation of the data and major conclusions of the study. These issues are related mainly to the lack of clarity in introducing the analysis strategy and in presenting the data. They are described in detail in Specific Comments.

Specific Comments

1.       It is not clear what data sets were used for the prediction analysis based on a classification tree approach as illustrated in Figure 8. According to paragraph 4.2 (Materials and Methods, page 11), a group of 26 animals were irradiated in present study at 13 Gy resulting in differential outcome, i.e. alive or dead. It is logically expected that this group would be the subject of prediction analysis. However, the total number of animals involved in the analysis, according to Figure 8, is 61 (32+5+22+2). Were animals irradiated at 10 Gy (all alive) involved in the analysis? Were data sets from previous referred study for 15 Gy irradiated animals (all dead) involved in the study? What was the size of each group in referred studies? Was a sample of virtual animals from artificial dataset described in section 2.5.2 involved in the analysis?

2.       As it follows from Figure 8 which illustrates the major outcome of the study, discrimination thresholds for the prediction were established for each of the effective predictors: % of lymphocytes (discrimination threshold 79.4), lung perfusion volume by MAA/CT (89.6) and % of monocytes (2.0). Although the basis for the choice of this set of predictors can be inferred from the content of the manuscript, it is not explained explicitly nowhere in the text. Moreover, it is not explained how the given discrimination values were obtained/calculated. Meanwhile, Figure 8 legend exactly duplicates the main text lines 207-214 instead of providing additional explanations for the figure.

3.       It is stated in Materials and Methods (paragraph 4.8, lines 431-432) and in Discussion (lines 236-237) that “The 15 Gy results were used as training sets for the current study”. In the context of developing a predictive model, a “training set” is a data set used to identify effective predictors and establish their discrimination thresholds, that can be prospectively used to predict outcome for the “test set”. This, however, is not a case for the present study. Moreover, as it follows from the data, presented in the study, the 15 Gy data set cannot be used in general case as a training set to predict survival after 13 Gy irradiation, as explained in the following point.

4.       As it follows from Figure 6, for majority (11 out 20) predictors, there is a difference (very likely statistically significant) between values for 15 Gy group (all dead) and 13 Gy dead group, thus indicating that the value of a predictor correlates rather with radiation dose than with survival outcome (alive or dead). This observation is not mentioned nor discussed in the manuscript. This observation is also well illustrated in Figure 7, from which it is obvious that for the same predictors (11 out of 20) subpopulations of 15 Gy group and 13 Gy dead group do not belong to the same statistical population. This observation makes meaningless to build logistic regression curves for these predictors based on combined data from different doses and prompts to concentrate on the search of predictors that correlate with outcome (dead or alive) and don’t correlate with radiation dose.

5.       It is not clear what is the purpose and approach for creating an artificial dataset. It is stated in section 2.5.1 that “The multivariate analyses were conducted using an artificial dataset with imputed values for the single-variable animals from the second set as described below”. This statement raises a few questions. First, the multivariate analysis usually assumes that there is a correlation/interaction between individual predictors. Since the authors used single-variable animal subsets to generate artificial dataset, no information would be available at all on the correlation of individual predictors/variables. Such a dataset can only be generated assuming independent variables. Second, as stated in section 2.5.2, from the artificial dataset, 11 animals treated with 0 Gy and 11 animals treated with 13 Gy were randomly selected. How was it possible from the dataset based on 15 Gy experiments (four single-variable subsets) to generate animals treated with 13 Gy?

6.       The data on the expression of miRNAs is labelled in Figures 4 and 5 as “Normalized Relative Expression”. The rationale for using both terms, “normalized” and “relative” is not clear. Was the data normalised and then relative values calculated? Or both terms do describe the same feature of the data? As described in section 4.6 (Materials and Methods), the expression for each target “was compared to the expression level of miR 191-5p (not affected by irradiation) in the same blood sample”. This procedure can be considered as normalization given that PCR rate might vary from sample to sample (all assuming multiplex PCR which however is not stated in the Material and Methods). However, it is stated further in section 4.6 that “Each value was then divided by the corresponding value for miR 150-5p in the same sample”. Does it mean that each value was first normalised to miR 191-5p and then divided by miR 150-5p to calculate “normalized relative expression”? Meanwhile, dividing by expression for miR 150-5p and claiming that this “would show more pronounced  change because of marked decrease in expression of miR 150-5p” (line 161) is misleading and wrong. It might result in the apparent increase in expression for a given marker while in reality it is not changed.   

Author Response

We would like to thank the reviewer for your time and efforts in reviewing our manuscript entitled “Biomarkers to predict lethal radiation injury to the rat lung”.  In response to your suggestions and comments we have made multiple revisions to the manuscript, expanding the analysis and removing results from discussion section, and modifying the figures. We feel these changes have improved the manuscript and we sincerely hope you find it suitable for publication.

General comments

The manuscript reports the results of the study that aimed to investigate and establish effective biomarkers of lethal lung injury after exposure to ionizing radiation. An ultimate objective of the study is to predict lethal outcome at early stages before manifestation of any symptoms thus allowing to apply countermeasures to improve survival. The subject of investigation presents a great interest for a broad audience of researchers, and its importance is undoubtful. The study is based on the extensive pool of experimental data obtained using relevant range of techniques. The authors claim the establishment of a combination of three biomarkers and the use of the classification tree approach for the prediction of lethal outcome. There are however a few issues that undermine the interpretation of the data and major conclusions of the study. These issues are related mainly to the lack of clarity in introducing the analysis strategy and in presenting the data. They are described in detail in Specific Comments.

Specific Comments

  1. It is not clear what data sets were used for the prediction analysis based on a classification tree approach as illustrated in Figure 8. According to paragraph 4.2 (Materials and Methods, page 11), a group of 26 animals were irradiated in present study at 13 Gy resulting in differential outcome, i.e., alive or dead. It is logically expected that this group would be the subject of prediction analysis. The reviewer is correct in this assumption.

However, the total number of animals involved in the analysis, according to Figure 8, is 61 (32+5+22+2). Were animals irradiated at 10 Gy (all alive) involved in the analysis? Were data sets from previous referred study for 15 Gy irradiated animals (all dead) involved in the study?  No. See below. What was the size of each group in referred studies? Was a sample of virtual animals from artificial dataset described in section 2.5.2 involved in the analysis? The prediction tree is based on the dataset that includes all the 39 animals for which we had full data (0 Gy, 10 Gy, and 13 Gy) and 22 virtual animals: 11 at 0 Gy and 11 at 15 Gy. We have added Table 1, page 11 in the Methods section for clarity.

  1. As it follows from Figure 8 which illustrates the major outcome of the study, discrimination thresholds for the prediction were established for each of the effective predictors: % of lymphocytes (discrimination threshold 79.4), lung perfusion volume by MAA/CT (89.6) and % of monocytes (2.0). Although the basis for the choice of this set of predictors can be inferred from the content of the manuscript, it is not explained explicitly nowhere in the text. Moreover, it is not explained how the given discrimination values were obtained/calculated. Meanwhile, Figure 8 legend exactly duplicates the main text lines 207-214 instead of providing additional explanations for the figure.

Thank you. We have added this explanation to the manuscript (section 2.5.3) “CART uses a recursive binary splitting algorithm to build the classification tree. Using the entire data set, the cutoff for the optimal split for each variable is determined by minimizing the Gini index, which measures the “impurity” of the nodes created by the split. The variable with the lowest Gini index is selected for the split. This process is repeated in the two subsets defined by the first split, and all subsequent splits until there is no improvement in the classification, or the subsets become too small (we did not split nodes with fewer than 20 observations).”

The text in the legend of Figure 8 and manuscript have been modified.

  1. It is stated in Materials and Methods (paragraph 4.8, lines 431-432) and in Discussion (lines 236-237) that “The 15 Gy results were used as training sets for the current study”. In the context of developing a predictive model, a “training set” is a data set used to identify effective predictors and establish their discrimination thresholds, that can be prospectively used to predict outcome for the “test set”. This, however, is not a case for the present study. Moreover, as it follows from the data, presented in the study, the 15 Gy data set cannot be used in general case as a training set to predict survival after 13 Gy irradiation, as explained in the following point.

We appreciate the reviewer's point about the terms “training” and “test” set.  We have changed the text of the manuscript to indicate that we performed the experiment in two stages: first only 0 Gy and 15 Gy to select candidate biomarkers, and then examined the performance of the candidate biomarkers on rats receiving an intermediate dose of radiation where we collected data on all of them. See page Material and Methods Table 1.

All references to training sets have been removed.

  1. As it follows from Figure 6, for majority (11 out 20) predictors, there is a difference (very likely statistically significant) between values for 15 Gy group (all dead) and 13 Gy dead group, thus indicating that the value of a predictor correlates rather with radiation dose than with survival outcome (alive or dead). This observation is not mentioned nor discussed in the manuscript. This observation is also well illustrated in Figure 7, from which it is obvious that for the same predictors (11 out of 20) subpopulations of 15 Gy group and 13 Gy dead group do not belong to the same statistical population. This observation makes meaningless to build logistic regression curves for these predictors based on combined data from different doses and prompts to concentrate on the search of predictors that correlate with outcome (dead or alive) and don’t correlate with radiation dose.

The reviewer is correct that in figure 6 and 7, the majority of predictors correlate to radiation dose. In fact, higher doses of radiations result in more extreme values of a predictor, and that should be expected if the predictor correlates with radiation damage. However, it is important that the predictor works after adjusting for radiation dose, i.e. within the 13Gy group, where both outcomes are observed. Our intention was that the logistic regression curves serve to facilitate visualization of this relationship. 

Below are the more detailed results.

##               

##                 Alive Dead

##   0Gy,\n alive     18    0

##   10Gy,\n alive     6    0

##   13Gy,\n alive     8    5

##   13Gy,\n dead      2   11

##   15Gy,\n dead      0   11

## Confusion Matrix and Statistics

##

##          Observed

## Predicted Alive Dead

##     Alive     8    2

##     Dead      5   11

##                                          

##                Accuracy : 0.7308         

##                  95% CI : (0.5221, 0.8843)

##     No Information Rate : 0.5             

##     P-Value [Acc > NIR] : 0.01448        

##                                          

##                   Kappa : 0.4615         

##                                          

##  Mcnemar's Test P-Value : 0.44969        

##                                          

##             Sensitivity : 0.8462         

##             Specificity : 0.6154         

##          Pos Pred Value : 0.6875         

##          Neg Pred Value : 0.8000          

##              Prevalence : 0.5000         

##          Detection Rate : 0.4231         

##    Detection Prevalence : 0.6154         

##       Balanced Accuracy : 0.7308         

##                                          

##        'Positive' Class : Dead           

##

  1. It is not clear what is the purpose and approach for creating an artificial dataset. It is stated in section 2.5.1 that “The multivariate analyses were conducted using an artificial dataset with imputed values for the single-variable animals from the second set as described below”. This statement raises a few questions. First, the multivariate analysis usually assumes that there is a correlation/interaction between individual predictors. Since the authors used single-variable animal subsets to generate artificial dataset, no information would be available at all on the correlation of individual predictors/variables. Such a dataset can only be generated assuming independent variables. Second, as stated in section 2.5.2, from the artificial dataset, 11 animals treated with 0 Gy and 11 animals treated with 13 Gy were randomly selected. How was it possible from the dataset based on 15 Gy experiments (four single-variable subsets) to generate animals treated with 13 Gy?

Point 1: The reviewer correctly identifies that we have no information regarding the correlation or interaction of the individual predictors.  We have added that limitation to the text of the manuscript (section 2.5.3).  Our purpose in showing data from the artificial dataset was to help readers visualize changes in individual variables (miRNA, lymphocytes, monocytes, etc.) at 60 days after different doses of irradiation, stratified by survival (on the Y axis).  Some variables increase with survival (e.g., lymphocytes and log2mi150-5p) while others increase with probability of death (e.g., duramycin or miRNA144-3p). This depiction is meant to help readers understand why a CART approach with multiple variables will ultimately be helpful in distinguishing subjects destined to fail versus those highly likely to do well without intervention.  We have added the following text to Figure 7 legend “These data show anticipated directional changes in individual variables stratified by survival status.”

Point 2: Our text on (section 2.5.2) should have said “11 animals treated with 0 Gy and 11 animals treated with 15 Gy” were selected. Some 0 Gy, all the 10 Gy and 13 Gy animals had complete data, and they were included as is. The text is now corrected.

If the reviewer still feels strongly that the data in the figure do not add significantly to the paper, we can remove them.

  1. The data on the expression of miRNAs is labelled in Figures 4 and 5 as “Normalized Relative Expression”. The rationale for using both terms, “normalized” and “relative” is not clear. Was the data normalised and then relative values calculated? Or both terms do describe the same feature of the data? As described in section 4.6 (Materials and Methods), the expression for each target “was compared to the expression level of miR 191-5p (not affected by irradiation) in the same blood sample”. This procedure can be considered as normalization given that PCR rate might vary from sample to sample (all assuming multiplex PCR which however is not stated in the Material and Methods). However, it is stated further in section 4.6 that “Each value was then divided by the corresponding value for miR 150-5p in the same sample”. Does it mean that each value was first normalised to miR 191-5p and then divided by miR 150-5p to calculate “normalized relative expression”? Meanwhile, dividing by expression for miR 150-5p and claiming that this “would show more pronounced change because of marked decrease in expression of miR 150-5p” (line 161) is misleading and wrong. It might result in the apparent increase in expression for a given marker while in reality it is not changed.

Responses:

The data on the expression of miRNAs is labelled in Figures 4 and 5 as “Normalized Relative Expression”. The rationale for using both terms, “normalized” and “relative” is not clear. Was the data normalised and then relative values calculated? Or both terms do describe the same feature of the data?

The data on the expression of miRNAs is labelled in Figures 4 and 5 as “Normalized Relative Expression”. The rationale for using both terms, “normalized” and “relative” is not clear. Was the data normalised and then relative values calculated? Or both terms do describe the same feature of the data?

In figure 4, the data of each individual microRNA were normalized (divided) by that of miR191-5p, while in figure 5, the data were normalized by miR150-5p. We agree with the reviewer that the titles of the y axis of these figures are confusing. We have changed the y axis of figures 4 and 5 as “Relative Expression”. 

As described in section 4.6 (Materials and Methods), the expression for each target “was compared to the expression level of miR 191-5p (not affected by irradiation) in the same blood sample”. This procedure can be considered as normalization given that PCR rate might vary from sample to sample (all assuming multiplex PCR which however is not stated in the Material and Methods).

We agree with the reviewer’s point that normalization may not be necessary in many cases.  Normalization was performed in this study to serve as a way to control for or reduce the possible variation in RT as well as PCR due to human errors, eg. non-consistent amount of total RNA or reagents used among samples, non-accurate cDNA dilution factor on specific samples, uneven heating temperatures among wells.

Specific reference standards unaffected by radiation are also controversial. Usually GAPDH and β-actin are used for mRNAs, but they are not appropriate to be used as a reference for miRNAs as well as for radiation models. A universally accepted reference for microRNA after radiation is not available. We have emphasized in the manuscript (section 2.4) that the expression level of miR 191-5p was not affected by irradiation from our previous publications, therefore was selected a good reference in our model. To avoid confusion, miR191-5p is clearly labeled in subtitles of each microRNA in figure 4.

However, it is stated further in section 4.6 that “Each value was then divided by the corresponding value for miR 150-5p in the same sample”. Does it mean that each value was first normalised to miR 191-5p and then divided by miR 150-5p to calculate “normalized relative expression”?

The data were only normalized to miR150-5p in figure 5. We have modified the manuscript in section 4.6 from “Each value was then divided by the corresponding value for miR 150-5p in the same sample” to “Each value was then divided by the corresponding value for miR 150-5p in the same sample instead of by miR 191-5p”. 

Meanwhile, dividing by expression for miR 150-5p and claiming that this “would show more pronounced change because of marked decrease in expression of miR 150-5p” (line 161) is misleading and wrong. It might result in the apparent increase in expression for a given marker while in reality it is not changed. 

We appreciate the reviewer’s point. See our response to Reviewer 1, question 2 as well. The miR150-5p is the only down regulated microRNA we found. A ratio of two microRNAs that change in opposite directions after radiation increases the sensitivity of miRNAs to detect injury evoked changes. Normalization to another miRNA also serves as a way to control/reduce the possible variation on RT as well as PCR due to human errors, eg. non-consistent amount of total RNA or reagents used among samples, non-accurate cDNA dilution factor on specific samples, uneven heating temperatures among wells.

Accordingly, we have modified the manuscript in section 2.4 to say:  “The ratio of the expression of each of the 5 miRNAs to miR 150-5p was also calculated for each marker (Figures 5A-E). The products were shown as a relative expression, which is a combination of two oppositely changed microRNA to increase sensitivity of radiation-evoked changes in miRNA expression and to normalize for variability based on pipetting, etc. This ratio was higher for miR19a-3p/150-5p, miR142-5p/150-5p, miR144-5p/150-5p and miR21-5p/150-5p after 13 Gy WTLI. The ratio of expression of miR144-5p/150-5p and miR 21-5p/150-5p were also increased at 2 weeks after 10 Gy WTLI (Figure 5E).    

In closing, the authors appreciate the efforts of the reviewers and editors’ suggestions that improved this manuscript.

Thank you again,

Round 2

Reviewer 1 Report

The revised version is marginally improved.

Comments:

1.     The authors’ probably have data (lung perfusion, WBC, neutrophils, lymphocytes, and monocytes) for individual rats from various time points till their survival or death ( 0 – 100 days and beyond). Do changes in lung perfusion, lymphocyte and monocyte percentages, and mortality/survival agree with predictions? This is important and should be discussed.

2.     In the methods section, include criteria for assessment of morbidity.

3.     I think normalizing miRNAs, that have already been normalized to miR 191-5p (remains constant with treatment), to miR 150-5p (decreases with treatment) is not correct. Further, according to the data presented miRNA change is not a biomarker for prediction of radiation lethality. Based on these points, the miRNA data is redundant and should be removed. Were miRNAs (Figs 4 and 5) measured after 2 weeks following radiation

4.     The results of imputation method (Figures 6 and 7) to predict survival should be presented as supplementary figures. 

5.     The discussion section is not improved in response to my comments.

Author Response

The revised version is marginally improved.

We thank the Reviewer for the care to clarify and further improve the manuscript. We have responded to each comment below, and made changes to the original manuscript as needed (see revised version with tracked changes). We hope our responses are satisfactory. 

Comments: 

  1. The authors’ probably have data (lung perfusion, WBC, neutrophils, lymphocytes, and monocytes) for individual rats from various time points till their survival or death ( 0 – 100 days and beyond). Do changes in lung perfusion, lymphocyte and monocyte percentages, and mortality/survival agree with predictions? This is important and should be discussed.

Results of changes in pulmonary perfusion [44], apoptotic injury [44], miRNA ex-pression [46] and circulating blood cell counts [46] were already established at 1-, 2-, 3- and 4-weeks post-15 Gy WTLI. For the survival studies reported in this manuscript, we used the earliest time point only for data collection and then followed the rats to survival making very limited measurements in order to minimize handling (eg blood collection) that could affect survival.  For this reason, we do not have data from additional time points for the cohort of rats reported here.

  1. In the methods section, include criteria for assessment of morbidity.

The rats were evaluated for euthanasia using criteria listed here. Single criterion for euthanasia includes open mouthed breathing, comatose state, seizures, or a BUN >120 mg dL-1. Multiple criteria e.g., are also considered body weight loss, inactivity when stimulated, hunched posture, increased respiratory effort and firm distended thorax for determination of euthanasia. The terminal endpoint for such rats was necropsy.

These criteria were deliberately not listed in our manuscript because this lab has been targeted by animal rights groups based upon the criteria in previous publications over 8 years ago (e.g. Medhora, 2015 below).  We have discussed our criteria with other groups in the same NIAID countermeasure program and are consistent with the consensus reached. But we are most reluctant to open ourselves to the security risks associated with such a response again.

Please seriously consider our request to list the reference [23] below which does detail the criteria for sacrifice as sufficient.

Medhora, M.; Gao, F.; Glisch, C.; Narayanan, J.; Sharma, A.; Harmann, L.M.; Lawlor, M.; Snyder, L.A.; Fish, B.L.; Down, J.D.; et al. Whole-thorax irradiation induces hypoxic respiratory failure, pleural effusions and cardiac remodeling. Journal of Radiation Research 2015, 56, 248-260, doi:10.1093/jrr/rru095.

  1. I think normalizing miRNAs, that have already been normalized to miR 191-5p (remains constant with treatment), to miR 150-5p (decreases with treatment) is not correct. Further, according to the data presented miRNA change is not a biomarker for prediction of radiation lethality. Based on these points, the miRNA data is redundant and should be removed. Were miRNAs (Figs 4 and 5) measured after 2 weeks following radiation?.

We appreciate the reviewer’s point that miRNA do not predict radiation lethality.  Nevertheless miRNA have been used widely as markers of lung conditions (PMID: 24066954) and even radiation sensitivity (PMID: 20728239).  The fact that no single miRNA emerged as a marker may be helpful to future investigators.

To meet the reviewer’s concern, we propose that we remove Figure 5 which normalizes other miRNA to the only miRNA we found which decreased with treatment.  We prefer to keep figure 4 with the intention that it may provide guidance for other investigators studying radiation induced changes in miRNA from circulating cells.

If the reviewer or editors feel strongly that it is a distraction, these data could be moved to the supplemental figures.  

Yes, miRNAs were measured 2 weeks following radiation. See lines 86.

  1. The results of imputation method (Figures 6 and 7) to predict survival should be presented as supplementary figures. 

We have moved figures 6 and 7 (now figures 5 and 6 due to removal of previous figure 5; see above) to the supplementary figures.

  1. The discussion section is not improved in response to my comments.

Please see the attachment which detail changes in the discussion section from the first submission to this last submission.  The discussion is substantially truncated and rewritten.

Thank you again for your careful review.

Round 3

Reviewer 1 Report

The authors' have addressed my concerns and comments satisfactorily.